# UNLOCKING EXPLORATION IN RLVR: UNCERTAINTY-AWARE ADVANTAGE SHAPING FOR DEEPER REASONING

## ABSTRACT

Reinforcement Learning with Verifiable Rewards (RLVR) has shown significant promise for enhancing the reasoning capabilities of large language models (LLMs). However, prevailing algorithms like GRPO broadcast a uniform advantage signal across all tokens in a sequence. This coarse-grained approach overlooks the pivotal role of uncertain, high-stakes decisions during reasoning, leading to inefficient exploration and the well-documented problem of entropy collapse. To address this, we introduce **Un**Certainty-aware **A**dvantage **S**haping (**UCAS**), a model-free method that refines credit assignment by leveraging the model's internal uncertainty signals. UCAS operates in two stages: it first modulates the response-level advantage using the model's overall self-confidence, and then applies a token-level penalty based on raw logit certainty. This dual mechanism encourages exploration of high-uncertainty paths that yield correct answers while penalizing overconfident yet erroneous reasoning, effectively balancing the exploration-exploitation trade-off. Extensive experiments on five mathematical reasoning benchmarks show that UCAS significantly outperforms strong RLVR baselines across multiple model scales, including 1.5B and 7B. Our analysis confirms that UCAS not only achieves higher rewards but also promotes greater reasoning diversity and successfully mitigates entropy collapse.

## 1 INTRODUCTION

Reinforcement learning (RL) has recently become a cornerstone for enhancing the complex reasoning abilities of Large Language Models (LLMs), moving beyond simple pattern matching toward more robust problem-solving. Among the various RL approaches, Reinforcement Learning with Verifiable Rewards (RLVR) has proven particularly effective. In this paradigm, a policy model explores a vast solution space and receives feedback from verifiable signals, such as the correctness of a final answer in mathematical reasoning. This direct feedback loop has enabled policy optimization algorithms like Group Relative Policy Optimization (GRPO) (Shao et al., 2024) to achieve substantial performance gains, powering state-of-the-art systems such as DeepSeek-R1 (Guo et al., 2025).

However, the success of RLVR reveals a critical underlying tension: the trade-off between precision and diversity. While methods like GRPO excel at increasing the probability of generating correct answers, they often do so at the cost of exploration. Due to the absence of a critic model, the learning signal in GRPO, which applies a single uniform advantage across all tokens, provides an indiscriminate and overly coarse form of credit assignment. It rewards all steps of a correct path equally and penalizes all steps of an incorrect one, failing to distinguish crucial reasoning leaps from trivial ones. This coarse-grained feedback drives the policy to converge prematurely on a small set of "safe" high-reward trajectories. A common side effect is **entropy collapse** (Cui et al., 2025b), where the output distribution contracts, reducing solution diversity and impairing performance on complex problems that demand novel reasoning strategies.

Previous studies (Wang et al., 2023; Lightman et al., 2024; Chen et al., 2024; Zhang et al., 2024; Sun et al., 2025a) have attempted to employ process-level reward models to deliver more fine-grained signals. However, as DeepSeek (Guo et al., 2025) points out, training fine-grained reward models is

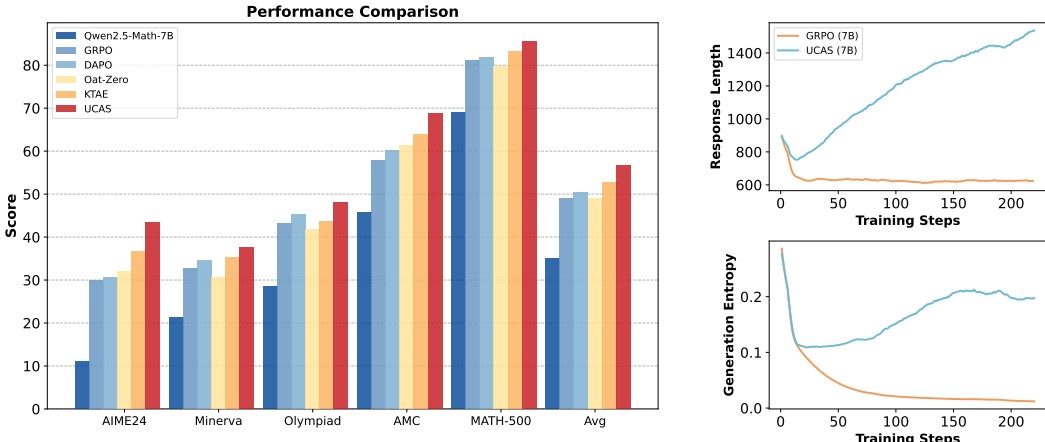

Figure 1: *Left*: Benchmark results across five math reasoning datasets, where our UCAS consistently outperforms RLVR baselines trained on models of the same parameter scale. *Right*: Training trajectories of UCAS and GRPO on Qwen2.5-Math-7B, showing that UCAS experiences an initial decline but subsequently rises in response length and generation entropy as training progresses. In contrast, GRPO exhibits a continual downward trend in entropy, reflecting the phenomenon of entropy collapse.

costly, difficult to scale, limited in its ability to provide accurate signals, and vulnerable to reward hacking. Some recent efforts (Chen et al., 2025; Cheng et al., 2025; Wang et al., 2025a) have tried to incorporate entropy-based feedback to enhance advantages, such as integrating semantic entropy or policy entropy related to the response into advantage calculations. Yet, most studies either pursue low entropy to improve accuracy or encourage high entropy to maintain exploration, lacking fine-grained modeling of the relationship between responses and their policy entropy.

To address the above challenges, we propose an **UnCertainty-aware Advantage Shaping (UCAS)**, a model-free method that refines credit assignment in RLVR by leveraging the model's intrinsic uncertainty. UCAS is designed to resolve the precision–diversity dilemma by reshaping the advantage signal at two complementary levels. At the **response level**, UCAS modulates the sequence-level advantage using the model's overall self-confidence, amplifying rewards for correct-but-uncertain responses and penalties for incorrect-but-confident ones. At the **token level**, it further introduces a certainty-based penalty derived directly from raw logits, discouraging local overconfidence while preserving diversity in reasoning. Collectively, these mechanisms promote exploration of uncertain but potentially fruitful reasoning paths, while efficiently suppressing confidently wrong solutions. Extensive experiments on five mathematical reasoning benchmarks demonstrate that UCAS consistently outperforms strong RLVR baselines at both the 1.5B and 7B model scales. Beyond reward improvements, UCAS fosters greater reasoning diversity and substantially mitigates entropy collapse, confirming the effectiveness of uncertainty as a fine-grained learning signal.

Our contributions can be summarized as follows:

- We propose UCAS, an extra-model-free fine-grained advantage shaping mechanism based on internal confidence signals, which performs uncertainty-aware advantage adjustment at both response and token levels.

- We provide a novel mechanism to adaptively calibrate advantages based on uncertainty, enabling steady reward gains, longer reasoning chains, and entropy recovery, thus preventing entropy collapse in RLVR and improving reasoning accuracy.

- Extensive experiments on multiple mathematical reasoning benchmarks demonstrate that UCAS significantly improves model reasoning performance, validating its effectiveness in enhancing exploration diversity and optimization outcomes.

## 2 BACKGROUND: REINFORCEMENT LEARNING WITH VERIFIABLE REWARDS

In the training of large language models, early mainstream reinforcement learning alignment methods primarily relied on PPO. By introducing a clipping ratio into the objective function, PPO stabilizes training by constraining the magnitude of policy updates. This method has been widely adopted in Reinforcement Learning from Human Feedback (RLHF), where reward models provide preference-based scores to gradually shape model behavior. However, PPO exhibits key limitations: it depends on critic-based value estimation and requires large-scale preference annotation, both of which are costly and prone to noise accumulation.

To overcome these limitations, recent research has introduced RLVR. RLVR converts open-ended outputs into programmatically checkable signals, such as numerical consistency in mathematics, unit-test pass rates in code generation, or formal constraint satisfaction (Su et al., 2025; Wang et al., 2025b), thereby avoiding the noise and cost of preference models. By forming a closed loop of model–environment–verifier, RLVR enables policies to be updated directly from binary or graded correctness signals, improving both sample efficiency and reproducibility in structured reasoning tasks.

In the concrete implementation of RLVR, GRPO (Shao et al., 2024) emerges as a representative algorithm. Unlike PPO, which relies on critic-based value estimation, GRPO computes advantages by normalizing group-level verifiable rewards and updates the policy directly.

Formally, the objective is given by:

$$\mathcal{J}_{\text{GRPO}}(\theta) = \mathbb{E}_{q \sim \mathcal{D}, \, o \sim \pi_{\theta_{\text{old}}}}$$

$$\left[ \frac{1}{G} \sum_{i=1}^{G} \frac{1}{|o_i|} \sum_{t=1}^{|o_i|} \quad \min(r_{i,t}(\theta)\hat{A}_{i,t}, \text{clip}(r_{i,t}(\theta), 1-\epsilon, 1+\epsilon)\hat{A}_{i,t}) - \beta D_{\text{KL}}(\pi_\theta \| \pi_{\text{ref}}) \right] \tag{1}$$

where

$$r_{i,t}(\theta) = \frac{\pi_\theta(o_{i,t} \mid q, o_{i,<t})}{\pi_{\theta_{\text{old}}}(o_{i,t} \mid q, o_{i,<t})}, \tag{2}$$

denotes the probability ratio between the new and old policies for token $o_{i,t}$, and the advantage $\hat{A}_{i,t}$ is estimated from group rewards as:

$$\hat{A}_{i,t} = \frac{R_i - \mu(R)}{\sigma(R) + \epsilon}, \tag{3}$$

with $R_i$ the cumulative verifiable reward of trajectory $o_i$, $\mu(R)$ and $\sigma(R)$ the mean and standard deviation across the sampled group, and $\epsilon$ a small constant for numerical stability.

By eliminating dependency on value models and instead exploiting group-normalized verifiable rewards, GRPO achieves stable and cost-efficient training.

Building on GRPO, Decouple Clip and Dynamic Sampling Policy Optimization (DAPO) (Yu et al., 2025) is proposed to further improve stability and exploration. DAPO integrates four key techniques: Clip-Higher, Dynamic Sampling, Token-Level Policy Gradient Loss, and Overlong Reward Shaping. Similar to GRPO, DAPO samples multiple responses per prompt and optimizes the following objective:

$$\mathcal{J}_{\text{DAPO}}(\theta) = \mathbb{E}_{(q,a) \sim \mathcal{D}, \, \{o_i\}_{i=1}^{G} \sim \pi_{\theta_{\text{old}}}(\cdot|q)}$$

$$\left[ \frac{1}{\sum_{i=1}^{G} |o_i|} \sum_{i=1}^{G} \sum_{t=1}^{|o_i|} \min\left( r_t^i(\theta)\hat{A}_t^i, \, \text{clip}\left(r_t^i(\theta), 1-\epsilon_{\text{low}}, \, 1+\epsilon_{\text{high}}\right)\hat{A}_t^i \right) \right], \tag{4}$$

$$\text{s.t.} \quad 0 < \left| \{i \in \{1, \ldots, G\} \mid \text{is\_equivalent}(o^i, a)\} \right| < G$$

where $\epsilon_{\text{low}}$ and $\epsilon_{\text{high}}$ denote the lower and upper bounds of the clipping range. Compared to GRPO, DAPO explicitly decouples the clipping bounds, incorporates adaptive sampling strategies, thereby alleviating entropy collapse and improving the generalizability of RLVR-trained models.

## 3 METHOD

To address the coarse credit assignment problem in RLVR, we introduce **Uncertainty-aware Advantage Shaping (UCAS)**, a method designed to replace the blunt instrument of uniform advantage with a more nuanced, two-stage mechanism. The central idea is to reshape the learning signal by considering uncertainty at two distinct granularities: the entire reasoning path (response-level) and the individual generative steps within it (token-level). This hierarchical approach first sets a *strategic* learning objective by evaluating the value of the overall trajectory, and then *locally* refines the policy update to encourage robust exploration and prevent the premature convergence that leads to entropy collapse.

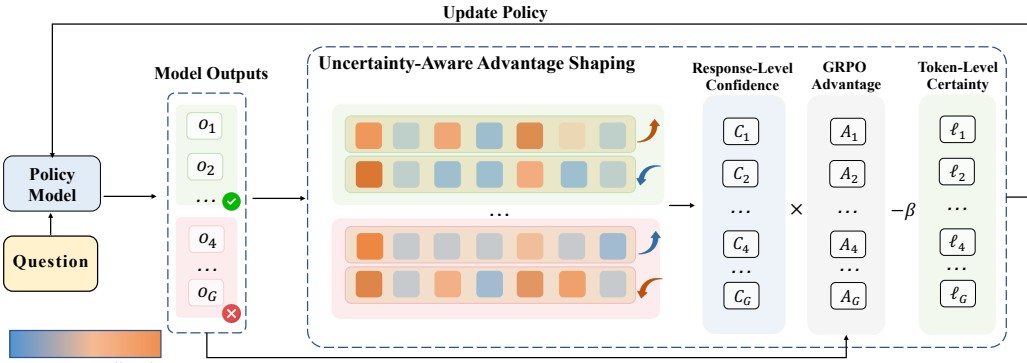

Figure 2: **Overview of the UCAS Advantage Shaping Mechanism.** UCAS refines the uniform GRPO advantage through a two-stage process. **Stage 1 (Macro-level)**: It applies Response-Level Modulation using the trajectory's overall self-confidence to determine its strategic value for exploration vs. exploitation. **Stage 2 (Micro-level)**: It introduces a Token-Level Certainty Penalty using raw logits to discourage local overconfidence. The final shaped advantage $\hat{A}_{i,t}^{\text{UCAS}}$ guides a more nuanced policy update.

### 3.1 UNCERTAINTY SIGNALS: FROM CONFIDENCE TO LOGITS

To perform this hierarchical shaping, UCAS requires signals that capture the model's epistemic state at both macro and micro levels. We extract these directly from the model's intrinsic generative process, avoiding the need for auxiliary networks.

**Response-Level Confidence.** For a high-level assessment of a full reasoning trajectory, we use the model's **self-confidence**. As defined in Kang et al. (2025), this is the average KL-divergence between the model's predictive distribution and a uniform distribution over the vocabulary $\mathcal{V}$. We denote this as $\mathcal{C}(o_i|q)$:

$$\mathcal{C}(o_i|q) := \frac{1}{|o_i|} \sum_{t=1}^{|o_i|} \text{KL}\big(U(\mathcal{V}) \,\|\, p_{\pi_\theta}(\cdot \mid q, o_{i,<t})\big) \tag{5}$$

A higher value of $\mathcal{C}(o_i|q)$ signifies higher overall confidence (low uncertainty) in the generated sequence, suggesting the model is following a well-trodden path.

**Token-Level Certainty.** While self-confidence is effective at the sequence level, it is derived from post-softmax probabilities, which can suffer from poor calibration (Liu et al., 2025a; Ma et al., 2025). This can cause the model to appear equally confident in different choices, masking subtle but important variations in uncertainty. To capture a more direct and sensitive signal at the token level, we use the model's raw **logit** value for the chosen token $o_{i,t}$ as a proxy for certainty. Let $\ell_{i,t}$ be the logit corresponding to token $o_{i,t}$ at step $t$. A higher logit value indicates greater model certainty in its choice, prior to softmax normalization.

## 3.2 UCAS: Two-Stage Advantage Shaping

Given a group of $G$ responses $\{o_1, \ldots, o_G\}$ to a prompt $q$, UCAS reshapes the original GRPO advantage $\hat{A}_i$ into a fine-grained, token-specific advantage $\hat{A}_{i,t}^{\text{UCAS}}$. This process unfolds in two complementary stages.

**Stage 1: Response-Level Advantage Modulation.** This stage adjusts the advantage for an entire response to encourage exploration of novel correct paths and suppress confident, well-trodden incorrect paths. First, we compute the self-confidence $\mathcal{C}(o_i|q)$ for each response $o_i$ in the group. To assess confidence relative to other responses in the same group, we apply z-score normalization:

$$\hat{\mathcal{C}}_i = \frac{\mathcal{C}(o_i|q) - \mu_{\mathcal{C}}}{\sigma_{\mathcal{C}} + \epsilon}, \tag{6}$$

where $\mu_{\mathcal{C}}$ and $\sigma_{\mathcal{C}}$ are the mean and standard deviation of confidence scores across the group.

We then compute a modulation weight $W(\hat{\mathcal{C}}_i)$ based on the sign of the original advantage $\hat{A}_i$, which directly encodes the correctness of the answer. Theoretically, we select an exponential form to act as a non-linear filter. This addresses the compressed variance often found in group-normalized scores, where linear rescaling fails to sufficiently distinguish "novel" exploration from "routine" exploitation.

$$W(\hat{\mathcal{C}}_i) = \begin{cases} \exp(-\alpha \cdot \hat{\mathcal{C}}_i) & \text{if } \hat{A}_i > 0 \quad \text{(Correct response)} \\ \exp(\alpha \cdot \hat{\mathcal{C}}_i) & \text{if } \hat{A}_i < 0 \quad \text{(Incorrect response)} \end{cases} \tag{7}$$

where $\alpha > 0$ is a hyperparameter controlling the shaping intensity. This formulation ensures that for correct responses, lower confidence (negative $\hat{\mathcal{C}}_i$) results in a larger weight, amplifying the reward. For incorrect responses, higher confidence (positive $\hat{\mathcal{C}}_i$) results in a larger weight, amplifying the penalty. The resulting modulated advantage is $\hat{A}_i^{\text{mod}} = W(\hat{\mathcal{C}}_i) \cdot \hat{A}_i$.

**Stage 2: Token-Level Certainty Penalty.** Response-level modulation sets a global learning objective for each trajectory, but this modulated advantage, $\hat{A}_i^{\text{mod}}$, is still a uniform signal broadcast to all tokens within that sequence. This alone is insufficient to prevent the model from developing localized overconfidence—a key driver of entropy collapse. The second stage therefore introduces a token-specific penalty to directly address this. By penalizing high-certainty choices at each step, we encourage the model to maintain a degree of epistemic humility, which preserves local exploration.

We use the raw logit $\ell_{i,t}$ as our certainty measure and apply Min-Max normalization within each sequence to create a standardized penalty score $\hat{\ell}_{i,t} \in [0, 1]$:

$$\hat{\ell}_{i,t} = \frac{\ell_{i,t} - \min_k(\ell_{i,k})}{\max_k(\ell_{i,k}) - \min_k(\ell_{i,k})} \tag{8}$$

A value of $\hat{\ell}_{i,t}$ close to 1 indicates high relative certainty for that token choice. This penalty acts as a regularizer, complementing the directional guidance from Stage 1.

**Final Advantage Shaping Formula.** By combining these two stages, UCAS creates a composite advantage signal that is both globally informed and locally sensitive. The final shaped advantage for each token is:

$$\hat{A}_{i,t}^{\text{UCAS}} = \underbrace{\hat{A}_i^{\text{mod}}}_{\text{Global Direction}} - \underbrace{\beta \cdot \hat{\ell}_{i,t}}_{\text{Local Penalty}} \tag{9}$$

where $\beta > 0$ is a hyperparameter controlling the penalty strength. This composite structure steers the model toward novel correct solutions (via $\hat{A}_i^{\text{mod}}$) while ensuring it traverses reasoning paths with a healthy degree of caution (via the penalty term), thereby mitigating entropy collapse and fostering more robust problem-solving abilities. This final advantage term then replaces the original advantage

in the RL objective:

$$\mathcal{J}_{\text{UCAS}}(\theta) = \mathbb{E}_{(q,a)\sim\mathcal{D}, \{o_i\}_{i=1}^G\sim\pi_{\theta_{\text{old}}}(\cdot|q)}$$

$$\left[ \frac{1}{\sum_{i=1}^G |o_i|} \sum_{i=1}^G \sum_{t=1}^{|o_i|} \min\left( r_t^i(\theta)\hat{A}_{i,t}^{\text{UCAS}}, \ \text{clip}\left(r_t^i(\theta), 1-\epsilon_{\text{low}}, \ 1+\epsilon_{\text{high}}\right)\hat{A}_{i,t}^{\text{UCAS}} \right) \right],$$

$$\text{s.t.} \quad 0 < \left| \{ i \in \{1, \ldots, G\} \mid \text{is\_equivalent}(o^i, a) \} \right| < G \tag{10}$$

The complete implementation process of UCAS is shown in Algorithm 1.

---

**Algorithm 1** Uncertainty-aware Advantage Shaping (UCAS)

---

**Input:** A group of $G$ responses $\{o_i\}_{i=1}^G$ sampled from policy $\pi_\theta$, their rule-based rewards $\{R_i\}_{i=1}^G$. Hyperparameters $\alpha$ and $\beta$.
1: Compute standard group-normalized advantages $\{\hat{A}_i\}_{i=1}^G$ from $\{R_i\}_{i=1}^G$.
2: Compute response-level self-confidence $\{\mathcal{C}(o_i|q)\}_{i=1}^G$ for all responses.
3: Normalize confidences across the group to get $\{\hat{\mathcal{C}}_i\}_{i=1}^G$.
4: **for** $i = 1$ to $G$ **do** $\qquad\qquad\qquad\qquad\qquad$ ▷ **Stage 1: Response-Level Advantage Modulation**
5: $\quad$ Compute modulation weight $W(\hat{\mathcal{C}}_i)$ based on $\hat{A}_i$ using Eq. 7.
6: $\quad$ Modulate the advantage: $\hat{A}_i^{\text{mod}} \leftarrow W(\hat{\mathcal{C}}_i) \cdot \hat{A}_i$.
$\qquad\qquad\qquad\qquad\qquad\qquad\qquad\qquad$ ▷ **Stage 2: Token-Level Certainty Penalty**
7: $\quad$ Get the sequence of logits $\{\ell_{i,t}\}_{t=1}^{|o_i|}$ for the generated tokens in $o_i$.
8: $\quad$ Normalize logits within the sequence to get $\{\hat{\ell}_{i,t}\}_{t=1}^{|o_i|}$.
9: $\quad$ **for** $t = 1$ to $|o_i|$ **do**
10: $\qquad$ Compute the final UCAS advantage:
11: $\qquad$ $\hat{A}_{i,t}^{\text{UCAS}} \leftarrow \hat{A}_i^{\text{mod}} - \beta \cdot \hat{\ell}_{i,t}$.
12: $\quad$ **end for**
13: **end for**
**Output:** The set of token-level UCAS advantages $\{\hat{A}_{i,t}^{\text{UCAS}}\}$.

---

## 4 EXPERIMENTS

### 4.1 EXPERIMENTAL SETUP

**Training Data and Benchmarks.** During the training phase, we utilize the widely-used MATH dataset as our training set. To maintain consistency with prior research, we only use the more challenging subset of this dataset for training, specifically problems from levels 3 to 5. To comprehensively evaluate the reasoning capabilities of the model trained with our method, we select five widely recognized benchmarks in the mathematical reasoning domain for testing: AIME24 (LI et al., 2024), MATH-500 (Hendrycks et al., 2021), AMC (LI et al., 2024), Minerva (Lewkowycz et al., 2022), and OlympiadBench (Huang et al., 2024), which collectively contain 1,560 problems.

**Models and Baselines.** We employ two variants of the Qwen2.5-Math (Yang et al., 2024) series as our foundation models: Qwen2.5-Math-1.5B and Qwen2.5-Math-7B. First, to quantify the performance improvement introduced by our method, we select the widely used GRPO and DAPO algorithms as comparison baselines. Furthermore, to benchmark against existing reinforcement learning techniques, we also select the following representative methods for comparison:

- **Simple-RL-Zoo (Zeng et al., 2025)**: Based on Qwen2.5-Math-7B, trained on the math-level3-5 dataset using the standard GRPO algorithm with rule-based rewards.

- **PRIME-Zero (Cui et al., 2025a)**: An online PRM update approach that leverages implicit process rewards from rollouts and outcome labels without requiring explicit annotations.

- **OpenReasonerZero (Hu et al., 2025)**: A zero-RL baseline on Qwen2.5-7B employing the standard PPO algorithm for policy optimization.

- **Oat-Zero (Liu et al., 2025b)**: Built on Qwen2.5-Math-7B, trained with rule-based rewards using a modified Dr.GRPO algorithm that removes variance terms and applies token-level normalization in the policy loss.

- **GRPO with Entropy Adv. (Cheng et al., 2025)**: Extends RLVR training by incorporating a clipped, gradient-detached entropy term into the advantage function to encourage exploration.

- **KTAE (Sun et al., 2025b)**: A token-level advantage estimation method trained with DAPO, quantifying key-token contributions via statistical association tests and combining them with rollout-level advantages.

These baselines cover applications of fundamental RL algorithms, process-reward-based methods, and algorithms improved for specific tasks like mathematical reasoning, aiming to evaluate the effectiveness and novelty of our method from multiple perspectives.

**Implementation Details.** We adopt the VERL framework (Sheng et al., 2024) and train our model using the optimization objective defined in Eq. 10. During training, the model's maximum context length is set to 4096, with a maximum prompt length of 1024 and a maximum response length of 3072. The learning rate is fixed at $1 \times 10^{-6}$, and the training batch size is set to 512. For each prompt, we sample 16 rollouts with a sampling temperature of 1.0. For the DAPO baseline, we use clipping thresholds of $\epsilon_{\text{low}} = 0.2$ and $\epsilon_{\text{high}} = 0.28$. The KL penalty loss and entropy regularization loss are omitted from the objective function. The hyperparameters for our UCAS method, $\alpha$ and $\beta$, are set to 0.25 and 0.01, respectively. All experiments are conducted on 2 compute nodes, each equipped with 8 NVIDIA A800 80GB GPUs.

## 4.2 MAIN RESULTS

The greedy pass@1 performance comparison between 1.5B and 7B models across five mathematical reasoning benchmarks is presented in Table 1. We can clearly find that **the UCAS model achieved the highest performance across all five math reasoning benchmarks on both the 1.5B and 7B parameter scales.** Compared with the DAPO baseline, UCAS improves the average accuracy from 41.2 to 47.3 (+6.1) on Qwen2.5-Math-1.5B and from 50.5 to 56.7 (+6.2) on Qwen2.5-Math-7B. Beyond DAPO, UCAS also surpasses strong baselines such as KTAE and Oat-Zero, with pronounced gains on challenging benchmarks including AIME24, AMC, and OlympiadBench. These results highlight the robustness and scalability of uncertainty-aware advantage shaping, demonstrating consistent benefits across model sizes and diverse reasoning tasks.

## 4.3 ANALYSIS

### 4.3.1 ABLATION STUDY

The ablation comparison between response-level and token-level uncertainty modeling is presented in Table 2. We can clearly observe that **both response-level and token-level uncertainty bring consistent gains over the DAPO baseline**. Compared with the model trained with DAPO, incorporating response-level confidence increases the average score on Qwen2.5-Math-1.5B from 41.2 to 44.7 (+3.5%), while token-level uncertainty further raises it to 45.1 (+3.9%). A similar trend holds on the 7B model, where both variants surpass the DAPO baseline. Their integration in UCAS achieves the best performance, confirming that both signals are individually useful and jointly necessary.

### 4.3.2 TRAINING DYNAMICS

The training process highlights several key performance trends, as shown in Figure 3. Compared to vanilla GRPO, UCAS demonstrates a consistent increase in the inference reward on the MATH500 benchmark. Regarding the average response length, the inclusion of UCAS enables the model to generate longer reasoning chains, reflecting more comprehensive problem-solving (Guo et al., 2025; Cheng et al., 2025), while simultaneously improving accuracy. For generation entropy, UCAS shows an early decline but later recovers and stabilizes at a higher level, effectively avoiding the entropy collapse reported in prior work (Cui et al., 2025b). Notably, the model's reward continues to rise

| Models | AIME24 | MATH-500 | AMC | Minerva | Olympiad | Avg |
|---|---|---|---|---|---|---|
| *Qwen2.5-Math-1.5B* | | | | | | |
| Base Model | 7.3 | 61.8 | 43.4 | 15.1 | 28.4 | 31.2 |
| GRPO | 15.6 | 76.0 | 51.8 | 22.1 | 36.3 | 40.4 |
| DAPO | 16.7 | 77.6 | 47.0 | 25.7 | 39.0 | 41.2 |
| Oat-Zero(Liu et al., 2025b) | 20.0 | 74.4 | 50.6 | 23.9 | 37.0 | 41.2 |
| KTAE(Sun et al., 2025b) | 20.0 | 77.6 | 50.6 | 29.0 | 40.0 | 43.4 |
| SEED-GRPO(Chen et al., 2025) | 23.3 | 75.4 | 50.6 | 26.8 | 41.3 | 43.5 |
| UCAS | **23.3** | **80.6** | **59.0** | **31.6** | **42.1** | **47.3** |
| *Qwen2.5-Math-7B* | | | | | | |
| Base Model | 11.0 | 69.0 | 45.8 | 21.3 | 28.4 | 35.1 |
| GRPO | 30.0 | 81.0 | 57.8 | 32.7 | 43.2 | 48.9 |
| DAPO | 30.5 | 81.8 | 60.2 | 34.5 | 45.3 | 50.5 |
| PRIME-Zero (Cui et al., 2025a) | 23.3 | 82.2 | 57.8 | 36.0 | 39.9 | 47.8 |
| OpenReasonerZero (Hu et al., 2025) | 17.9 | 78.4 | 45.8 | 27.9 | 45.0 | 43.0 |
| Oat-Zero(Liu et al., 2025b) | 32.1 | 79.8 | 61.4 | 30.5 | 41.8 | 49.1 |
| Simple RL-Zero(Zeng et al., 2025) | 26.7 | 78.6 | 59.0 | 33.8 | 43.4 | 48.3 |
| GRPO with Entropy Adv. (Cheng et al., 2025)[†] | 33.7 | 83.1 | **69.8** | - | - | - |
| KTAE(Sun et al., 2025b) | 36.7 | 83.2 | 63.9 | 35.3 | 43.7 | 52.6 |
| SEED-GRPO(Chen et al., 2025) | 43.3 | 82.2 | 64.7 | 35.0 | 45.2 | 54.7 |
| UCAS | **43.3** | **85.6** | 68.7 | **37.6** | **48.0** | **56.7** |

Table 1: The greedy pass@1 performance of 1.5B and 7B models across five math reasoning benchmarks. †: results from Cheng et al. (2025). Our method UCAS consistently surpasses all baselines in both parameter scales.

| Models | AIME24 | MATH-500 | AMC | Minerva | Olympiad | Avg |
|---|---|---|---|---|---|---|
| *Qwen2.5-Math-1.5B* | | | | | | |
| Base Model | 7.3 | 61.8 | 43.4 | 15.1 | 28.4 | 31.2 |
| w/ DAPO | 16.7 | 77.6 | 47.0 | 25.7 | 39.0 | 41.2 |
| w/ DAPO + Response-Level Confidence | **23.3** | 79.6 | 51.8 | 27.6 | 41.0 | 44.7 |
| w/ DAPO + Token-Level Certainty | 20.0 | 80.2 | 55.4 | 29.7 | 40.1 | 45.1 |
| w/ DAPO + UCAS (Ours) | **23.3** | **80.6** | **59.0** | **31.6** | **42.1** | **47.3** |
| *Qwen2.5-Math-7B* | | | | | | |
| Base Model | 11.0 | 69.0 | 45.8 | 21.3 | 28.4 | 35.2 |
| w/ DAPO | 30.5 | 81.8 | 60.2 | 34.5 | 45.3 | 50.5 |
| w/ DAPO + Response-Level Confidence | 40.0 | 85.0 | 63.9 | 36.7 | 47.4 | 54.6 |
| w/ DAPO + Token-Level Certainty | 36.7 | 84.6 | 65.0 | 29.7 | 47.7 | 52.7 |
| w/ DAPO + UCAS (Ours) | **43.3** | **85.6** | **68.7** | **37.6** | **48.0** | **56.7** |

Table 2: Ablation study of uncertainty modeling. Both sentence-level and token-level uncertainty bring consistent gains over the DAPO baseline.

even as the entropy increases, which indicates a stable and effective training dynamic where exploration and optimization are well-balanced.

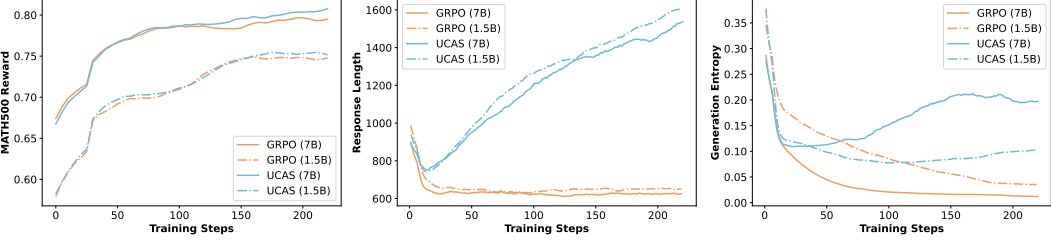

Figure 3: Training dynamics of UCAS compared with GRPO across both 7B and 1.5B models. **Left:** Reward; **Middle:** Response Length; **Right:** Generation Entropy.

### 4.3.3 CROSS-DOMAIN GENERALIZATION

To assess the generality of UCAS beyond mathematical reasoning, we conduct additional experiments on three diverse benchmarks: **LeetCode** (Guo et al., 2024) (code generation), **LiveCodeBench** (Jain et al., 2024) (competitive programming), and **MMLU** (Hendrycks et al., 2020) (general task reasoning). As shown in Table 3, despite being trained solely on mathematical reasoning data, UCAS consistently outperforms the strong DAPO baseline across all non-math tasks.

| Method | LeetCode (Pass@1) | LiveCodeBench (Pass@1) | MMLU (Acc) | Avg |
|---|---|---|---|---|
| Base Model | 11.7 | 5.7 | 65.7 | 27.7 |
| DAPO | 18.3 | 9.2 | 67.3 | 31.6 |
| UCAS (Ours) | **23.6** (+5.3) | **14.8** (+5.6) | **70.8** (+3.5) | **36.4** (+4.8) |

Table 3: Generalizing UCAS from math-only training to evaluations on non-math tasks.

This strong transferability suggests that our uncertainty-aware exploration mechanism is a broadly applicable principle. By unlocking exploration for high-uncertainty paths, UCAS improves performance not just in calculation but also in the multi-step logical planning required for programming and general reasoning, demonstrating gains beyond the mathematical domain.

### 4.3.4 PASS@K EVALUATION

Prior studies (Wang et al., 2022; Wu et al., 2024) have shown that with a limited number of rollouts, models often struggle to solve certain tasks. In contrast, when the rollout budget is sufficiently large, the probability of sampling effective solutions increases considerably. This observation suggests that pass@k accuracy with a large k provides a more reliable estimate of a model's potential performance (Yue et al., 2025). Under this evaluation protocol, a problem is considered solved if any of the k sampled reasoning trajectories yield the correct answer.

Figure 4 reports pass@k results on the AIME24 benchmark. The results indicate that UCAS achieves more consistent improvements as k grows. In contrast, Vanilla-GRPO and its enhanced variants show slower growth, consistent with findings from Yue

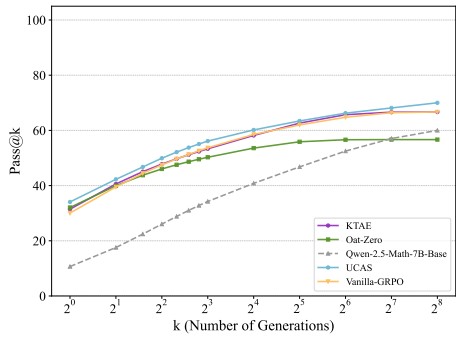

Figure 4: Comparison of pass@k results on the AIME24 Benchmark.

et al. (2025). The stronger performance of UCAS under the pass@k metric highlights its effectiveness, which can be attributed to differences in exploration strategies. Unlike Vanilla-GRPO, which often suffers from exploration stagnation, where the model repeatedly samples low-diversity rollouts, UCAS leverages uncertainty-aware advantage shaping to sustain diverse exploration and escape local optima.

## 5 RELATED WORK

### 5.1 RL FOR LLM REASONING

Recent advances in reinforcement learning have transformed the training of large language models for reasoning tasks. Process reward models (PRMs) (Lightman et al., 2023) have emerged as a key innovation, providing step-level supervision that improves both efficiency and accuracy compared to outcome-only rewards. Approaches such as PRIME (Cui et al., 2025a) eliminate costly human annotation by deriving implicit process feedback, while OmegaPRM (Luo et al., 2024) leverages Monte Carlo Tree Search (MCTS) to automatically identify reasoning errors. Alongside this, DeepSeek-R1 (Guo et al., 2025) demonstrates that sophisticated reasoning can emerge purely from RL without supervised fine-tuning, enabled by GRPO, which replaces value functions with group-based baselines.

These advances redefine alignment and reasoning in LLMs, positioning reinforcement learning with verifiable or process-level rewards as a scalable and principled alternative to preference-model-based RLHF.

## 5.2 REINFORCEMENT LEARNING FROM VERIFIABLE REWARDS

RLVR has emerged as a scalable alternative to preference-based alignment by converting open-ended outputs into checkable signals such as mathematical correctness or unit-test pass rates (Guo et al., 2025; Yue et al., 2025). While early implementations demonstrated strong gains in pass@1 accuracy, subsequent studies revealed a consistent *policy entropy collapse*: models rapidly concentrate probability mass on a narrow set of high-reward trajectories, diminishing output diversity and limiting exploration (Cui et al., 2025b). Empirical analyses show that RLVR-trained models often underperform base models on pass@$k$ (Shao et al., 2024; Yue et al., 2025), highlighting a precision–diversity trade-off (Wu et al., 2025; Dong et al., 2025).

Algorithmic responses to entropy collapse vary. Standard entropy or KL penalties provide partial remedies, though their effectiveness often depends heavily on the divergence form (Li et al., 2025). More recent uncertainty-aware approaches have sought to refine the learning signal, though with differing philosophies. For instance, SEED-GRPO (Chen et al., 2025) leverages semantic entropy to downscale updates for uncertain queries, adopting a conservative risk-mitigation strategy. In stark contrast, UCAS adopts an exploratory philosophy: we explicitly amplify rewards for correct-but-uncertain trajectories to incentivize venturing into novel reasoning domains, rather than inhibiting learning from uncertainty.

Similarly, while entropy-based shaping methods (Cheng et al., 2025) introduce indiscriminate entropy bonuses to encourage diversity, UCAS implements a *conditional*, two-stage mechanism. By combining response-level confidence with token-level raw logits which we find to be a more sensitive proxy for local overconfidence than post-softmax entropy, UCAS distinguishes between productive exploration and blind guessing. Unlike pure entropy-based frameworks, UCAS introduces correctness-contingent modulation, amplifying penalties for confident errors while guiding exploration through uncertainty, offering a more fine-grained solution to the entropy collapse problem than global regularization or token-level covariance control (Cui et al., 2025b).

## 6 CONCLUSION

In this work, we introduced UnCertainty-aware Advantage Shaping (UCAS), a fine-grained advantage estimation framework that leverages internal confidence signals without requiring additional reward models. By jointly modeling uncertainty at both the response and token levels, UCAS reshapes advantages to highlight critical uncertain reasoning steps and suppress overconfident yet erroneous segments. Experimental results on major mathematical reasoning benchmarks show that UCAS achieves substantial performance improvements over GRPO and its enhanced variants. Analysis of the training dynamics further reveals that, as training progresses, UCAS demonstrates steadily increasing rewards, longer reasoning chains, and an entropy trajectory that first declines and then rises, reflecting stronger exploratory capability. These findings indicate that uncertainty-aware advantage shaping offers an effective pathway toward more robust reinforcement learning for large language models.

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

## A   LIMITATION AND FUTURE WORK

While our work demonstrates the effectiveness of UCAS in the domain of mathematical reasoning, we acknowledge several limitations that present valuable opportunities for future research.

First, our experiments are exclusively focused on mathematical tasks, which benefit from clear, binary verifiable rewards (i.e., the answer is either correct or incorrect). The direct applicability of UCAS to domains with more nuanced, subjective, or dense reward signals, such as creative writing, summarization, or open-domain dialogue remains an open question. Adapting the uncertainty-aware shaping mechanism to these softer reward landscapes would be a crucial next step.

Second, our method relies on self-confidence and raw logits as proxies for model uncertainty. While these internal signals are computationally efficient and effective, future work could explore alternative or complementary uncertainty metrics. Techniques such as Monte Carlo dropout, model ensembles, or semantic entropy could potentially capture different facets of model uncertainty and lead to even more refined and robust advantage shaping. Investigating these areas will be essential for understanding the broader generalizability of our approach.

## B   IMPACT OF DIFFERENT HYPERPARAMETER WEIGHTS

To evaluate the robustness of UCAS and ensure the reliability of our findings, we conduct a sensitivity analysis on the two key hyperparameters: the response-level modulation coefficient $\alpha$ and the token-level penalty coefficient $\beta$. For each experiment, we vary one hyperparameter while fixing the other at its empirically optimal value, and report the post-training results of the Qwen-Math-1.5B model on the Math-500 benchmark.

Table 4: Sensitivity analysis on Math-500 when varying the token-level penalty $\beta$ (upper block) and the response-level modulation coefficient $\alpha$ (lower block).

| Response-Level $\alpha$ | Token-Level $\beta$ | Math-500 |
|:---:|:---:|:---:|
| *Varying Token-Level Penalty ($\beta$), Fixed $\alpha = 0.2$* | | |
| 0.2 | 0.005 | 79.6 |
| 0.2 | 0.01 | **80.4** |
| 0.2 | 0.05 | 78.8 |
| 0.2 | 0.1 | 78.4 |
| *Varying Response Modulation ($\alpha$), Fixed $\beta = 0.01$* | | |
| 0.1 | 0.01 | 80.0 |
| 0.2 | 0.01 | **80.4** |
| 0.4 | 0.01 | 78.8 |

**Impact of Token-Level Penalty ($\beta$).**   The upper block of Table 4 elucidates the critical trade-off controlled by the token-level penalty $\beta$. We observe that a moderate penalty ($\beta = 0.01$) yields optimal performance (80.4), suggesting it successfully balances the exploration-exploitation dynamic. Notably, when $\beta$ is too low (0.005), performance dips to 79.6, indicating that the penalty is insufficient to counteract the model's natural tendency toward overconfidence and entropy collapse. Conversely, as $\beta$ increases beyond 0.05, performance degrades further to 78.4. This suggests that excessive penalization over-regularizes the policy, forcing the model to artificially flatten its distribution even for necessary, high-certainty reasoning steps, thereby hindering the generation of coherent solution chains.

**Impact of Response-Level Modulation ($\alpha$).**   The lower block of Table 4 examines the sensitivity to the response-level modulation coefficient $\alpha$. The method exhibits stability within the range $\alpha \in [0.1, 0.2]$, where the modulation appropriately emphasizes uncertain-but-correct trajectories. However, increasing $\alpha$ to 0.4 results in a noticeable performance drop to 78.8. This deterioration

implies a "signal dominance" issue: when the modulation is overly aggressive, the confidence-based scaling factor begins to overshadow the fundamental correctness signal (the original advantage). This introduces high variance into the reward landscape, causing the policy optimization to drift away from the primary objective of mathematical accuracy in favor of gaming the uncertainty metric.

## C FURTHER ANALYSIS

To further analyze the effect of UCAS training, we compute the response-level confidence scores of model outputs according to Eq. 5, measured before and after training on Qwen2.5-Math-1.5B across MATH and Olympiad datasets. We focus on the MATH and Olympiad datasets because they contain more samples and a larger number of responses whose correctness changes after training, which makes them well suited for detailed analysis. For comparability, the confidence values are normalized by subtracting the mean and dividing by the standard deviation.

Based on the correctness of the responses before and after training, the samples are categorized into three groups: (i) consistently correct $(1{\rightarrow}1)$, (ii) correct before but incorrect after $(1{\rightarrow}0)$, (iii) incorrect before but correct after $(0{\rightarrow}1)$, and (iiii) incorrect before and incorrect after $(0{\rightarrow}0)$. Figure 5 illustrates the distribution of these categories, where each point represents model's response to a given problem.

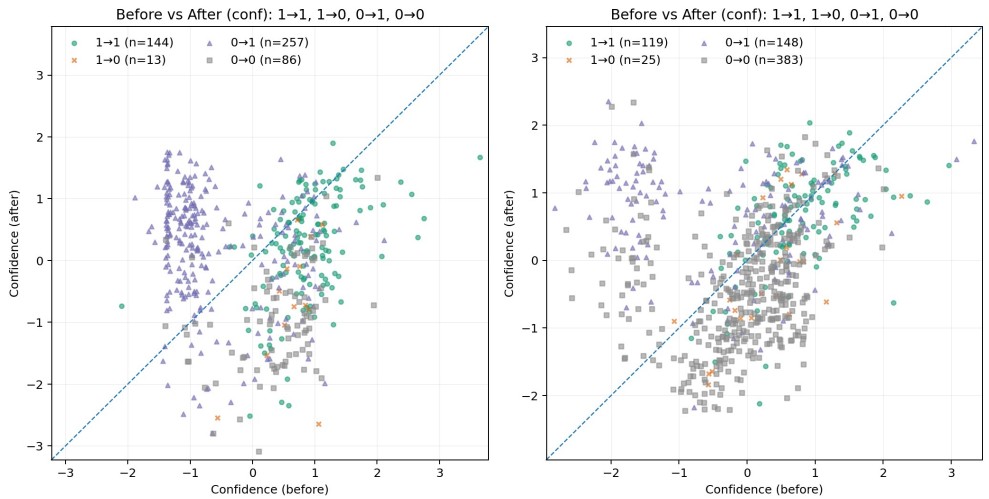

Figure 5: Confidence dynamics before and after UCAS training on the MATH and Olympiad datasets.

From Figure 5, we observe that for many problems correctly solved only after UCAS training $(0{\rightarrow}1)$, the model's confidence notably increases. In contrast, for problems that remain unsolved before and after training $(0{\rightarrow}0)$, the model tends to reduce its confidence, suggesting a more calibrated estimation of its own uncertainty.

## D LLMS USAGE STATEMENT

We employed a Large Language Model (LLM) to assist exclusively in the editorial stage of manuscript preparation. Its role was limited to refining phrasing, correcting grammar, and enhancing clarity and readability across different sections. The LLM had no involvement in formulating research ideas, designing experiments, or conducting analyses. All scientific contributions and findings are entirely the work of the authors. The authors have ensured that the use of the LLM complies with ethical standards, avoiding plagiarism and scientific misconduct.

