# OpenReview forum: "Unlocking Exploration in RLVR: Uncertainty-aware Advantage Shaping for Deeper Reasoning"
_ICLR.cc/2026/Conference — ICLR 2026 Conference Withdrawn Submission_

### Official Review · Reviewer_cqWa · 2025-10-15

**Soundness:** 3
**Presentation:** 3
**Contribution:** 1
**Rating:** 2
**Confidence:** 4

**Summary:**

This paper proposes UCAS (UnCertainty-Aware Advantage Shaping), a method to improve Reinforcement Learning with Verifiable Rewards (RLVR) for large language models (LLMs). The key idea is to shape the advantage signal using model uncertainty: (1) modulating the sequence-level advantage by the model’s self-confidence, and (2) penalizing token-level certainty derived from logits. The goal is to prevent entropy collapse and encourage exploration. The authors report improvements over GRPO and DAPO on several mathematical reasoning benchmarks (AIME24, MATH-500, AMC, Minerva, OlympiadBench) using Qwen2.5-Math-1.5B and 7B.

**Strengths:**

1. The paper tackles a real problem in RLVR: entropy collapse and insufficient exploration.

2. The proposed uncertainty-aware shaping is conceptually simple, easy to implement, and compatible with existing RLVR frameworks.

3. The presentation is clear, and the motivation is easy to follow.

**Weaknesses:**

###  Limited Novelty and Theoretical Justification

The core contributions lack sufficient novelty. The paper essentially applies existing uncertainty quantification techniques (self-confidence from prior work, raw logit values) to weight advantages differently. The two-stage mechanism is straightforward:

- Response-level: exponential weighting based on normalized confidence (Eq. 7)
- Token-level: min-max normalized logit penalty (Eq. 8)

Neither component introduces fundamentally new concepts. The exponential weighting scheme is ad-hoc without theoretical grounding for why this specific functional form is optimal. Why exponential rather than linear, polynomial, or other monotonic functions? The paper provides no principled justification beyond empirical performance.

### Weak conceptual contribution.

The paper presents UCAS as a new framework, but it does not introduce any fundamentally new algorithmic principle beyond applying confidence-based scaling to the GRPO advantage. Similar uncertainty-aware or entropy-regularized approaches already exist (e.g., semantic entropy regularization, variance-aware advantage estimation, or KTAE). The novelty over prior work is therefore marginal.

# Writing and Presentation Issues

The paper oversells the contribution. Terms like "unlocking exploration" and "deeper reasoning" are not substantiated by the actual improvements shown

The "entropy collapse" narrative is emphasized throughout, but Figure 3 shows UCAS entropy actually drops initially before recovering—this deserves more analysis

Some claims lack support: "encourages exploration of high-uncertainty paths that yield correct answers" (lines 25-26)—but the method equally amplifies penalties for wrong answers with high uncertainty

**Questions:**

How do results change with different α and β values?

Why does entropy initially drop before recovering (Figure 3)?

---

> ### Author Response · Authors · 2025-11-21
> **To Reviewer cqWa (1/3)**
>
> Thanks for your efforts to provide insightful comments. We will address your concerns point by point.
>
> > W1: Limited concept contribution
>
> Thanks for your comments. We respectfully disagree with the assessment that our contribution is marginal. While we utilize established uncertainty proxies (e.g. self-confidence, logits), our core novelty lies in the **conditional advantage shaping mechanism** and its specific **exploratory philosophy**, which fundamentally differs from prior other uncertainty-aware methods or KTAE.
>
> To clarify the unique position of UCAS, we provide a detailed comparison in `Table c1` against the methods mentioned by the reviewer (KTAE, Seed-grpo) and standard Entropy Regularization.
>
> **Table c1: Comparison of UCAS with related Advantage/Entropy shaping methods.**
>
> | **Method**                  | **UCAS (Ours)**                                              | **KTAE [1]**                                            | **Seed-grpo [2]**                                            | **Entropy Regularization**                         |
> | --------------------------- | ------------------------------------------------------------ | ------------------------------------------------------- | ------------------------------------------------------------ | -------------------------------------------------- |
> | **Core Signal**             | **Intrinsic Uncertainty** (KL + Logits)                      | **Statistical Frequency** (Fisher Test)                 | **Semantic Entropy** (Extra Clustering)                      | **Policy Entropy**                                 |
> | **Granularity**             | **Hierarchical** (Response + Token)                          | Token-Level Only                                        | Response-Level Only                                          | Global-Level                                       |
> | **Optimization Philosophy** | **Exploratory:** *Amplify* learning for "Uncertain but Correct" paths. | **Attribution:** Credit tokens correlated with success. | **Conservative:** *Suppress* learning for "Uncertain" paths (risk aversion). | **Randomness:** Indiscriminate push for diversity. |
>
> Regarding the reviewer’s comment on the weak conceptual contribution  between our approach and uncertainty-aware or entropy-regularized approaches, we believe there are key differences that set our work apart. As shown in `Table c1`, UCAS introduces a distinct algorithmic principle:
>
> **1. Contrast with Seed-grpo (Philosophy of Action):** While Seed-grpo also uses uncertainty, its goal is **risk mitigation**. It assumes high uncertainty implies data quality issues and *downscales* the advantage to prevent overfitting. **In contrast, UCAS is designed for exploration.** We identify "high uncertainty + correct outcome" as the most valuable learning signal (e.g. the "Aha!" moment). Our exponential weighting (`Eq. 7 of the manuscript`) *upscales* the advantage of these trajectories . This is not a minor variation but a fundamentally different optimization objective: Seed-grpo suppresses all uncertainties; UCAS utilizes it for exploration according to the reward sign.
>
> **2. Contrast with KTAE (Source of Signal):** KTAE relies on **post-hoc statistical attribution** (counting token frequency across group rollouts). It identifies tokens *correlated* with success but ignores the model's *internal epistemic state*. **UCAS is intrinsic.** By using raw logits (Stage 2), we penalize tokens where the model is *overconfident* (high logits) but wrong, or locally stagnant . This allows UCAS to address **entropy collapse** directly at the generation level, which statistical counting (KTAE) cannot do.
>
> [1] Sun W, Yang W, Jian P, et al. “KTAE: A Model-Free Algorithm to Key-Tokens Advantage Estimation in Mathematical Reasoning.” arXiv preprint arXiv:2505.16826, 2025.
>
> [2] Chen M, Chen G, Wang W, et al. "Seed-grpo: Semantic entropy enhanced grpo for uncertainty-aware policy optimization." arXiv preprint arXiv:2505.12346, 2025.

---

> ### Author Response · Authors · 2025-11-21
> **To Reviewer cqWa (2/3)**
>
> > W2: More theoretical analysis
>
> From a theoretical perspective, standard entropy regularization provides a global, indiscriminate push for diversity. In contrast, UCAS introduces a **state-dependent exploration bonus** contingent on correctness. By injecting uncertainty directly into the advantage (`Eq. 7 of the manuscript`), we create a principled separation: uncertainty acts as a *bonus* when the trajectory is correct (encouraging exploration of new solutions) but when incorrect it acts as a **mitigator**, reducing the penalty for low-confidence errors to "forgive" failed exploration attempts. Conversely, high confidence triggers a stricter penalty, targeting and suppressing **overconfident errors** rather than punishing all mistakes equally.
>
> In addition to the mechanistic explanation, the design of our exponential form is not arbitrary; it is chosen to amplify sensitivity while remaining strictly positive. In GRPO, z-scores often exhibit compressed variance, meaning linear rescaling is insufficient to meaningfully differentiate between "routine" and "novel" solutions.
>
> The exponential modulation ($W = \exp(-\alpha \cdot \mathcal{C})$ for correct responses) acts as a non-linear amplifier that **disproportionately rewards low-confidence (novel) correct answers** while assigning relatively lower rewards to high-confidence (routine) ones. This mechanism is crucial for preventing entropy collapse: it explicitly incentivizes the model to explore uncertain but valid reasoning paths rather than converging prematurely on safe, memorized solutions. Conversely, for incorrect answers, the exponential form aggressively penalizes overconfidence. Our ablation study in `Table 2 of the manuscript` empirically confirms that this specific non-linear reshaping—prioritizing exploration in successful trajectories is a key driver of performance gains.

---

> ### Author Response · Authors · 2025-11-21
> **To Reviewer cqWa (3/3)**
>
> > W3&Q2: Writing and Presentation Issues
>
> We thank the reviewer for their detailed attention to our presentation and claims. We appreciate the opportunity to clarify the training dynamics and the mathematical formulation of our shaping mechanism.
>
> 1. **On "Overselling" and "Deeper Reasoning":** We acknowledge the reviewer's concern regarding the terminology. However, we respectfully submit that these terms describe specific, measurable phenomena observed in our experiments:
>
>    * **"Deeper Reasoning":** As shown in `Figure 3 (Middle) of the manuscript`, UCAS drives a sustained increase in average response length throughout training (reaching ~1400 tokens vs. ~600 for GRPO). This quantitative increase in chain length correlates with improved performance, which we interpret as the model engaging in more extensive reasoning steps to solve complex problems.
>
>    * **"Unlocking Exploration":** This refers to the **entropy recovery** phenomenon shown in `Figure 3 (Right) of the manuscript`. While baselines suffer from "entropy collapse" (continual decline), UCAS reverses this trend, actively expanding the policy's search space in later training stages.
>
>      To avoid ambiguity, we will ensure these terms are explicitly linked to these specific metric definitions in the revised version.
>
> 2. **On Entropy Dynamics** (`Figure 3 of the manuscript`): The reviewer correctly observes the initial entropy drop. This "U-shaped" trajectory is not a contradiction but a desirable feature of our method.
>
>    * **Phase 1 (Initial Drop):** The model rapidly discards low-quality, high-entropy noise (e.g., gibberish or obvious errors). This is necessary for basic alignment.
>
>    * **Phase 2 (Recovery):** Crucially, unlike GRPO which continues to collapse, UCAS *recovers* entropy. This inflection point demonstrates that after learning the basics, the uncertainty-aware shaping successfully kicks in to incentivize the exploration of diverse, valid reasoning paths.
>
> 3. **Clarification on "Penalties for Wrong Answers"**: We respectfully point out a misunderstanding regarding how UCAS handles incorrect answers. The reviewer stated that the method "equally amplifies penalties for wrong answers with high uncertainty." **This is not the case.** According to `Eq. 7 of the manuscript` , our modulation is **state-dependent**:
>
>    - **Correct Responses ($\hat{A} > 0$):** Weight $W = \exp(-\alpha \cdot \hat{\mathcal{C}})$. High uncertainty (negative $\hat{\mathcal{C}}$) leads to $W > 1$. **Result:** We *amplify* the reward for exploration.
>    - **Incorrect Responses ($\hat{A} < 0$):** Weight $W = \exp(\alpha \cdot \hat{\mathcal{C}})$. High uncertainty (negative $\hat{\mathcal{C}}$) leads to $W < 1$. **Result:** We *attenuate* (reduce) the penalty.
>
>    Therefore, UCAS does **not** amplify penalties for uncertain errors. Instead, it is **lenient towards uncertain failures** (treating them as "failed exploration attempts") while strictly penalizing **confident errors** (where $W > 1$). This asymmetry is precisely why we claim the method encourages exploration: it makes it "safer" for the model to try uncertain paths without fearing a catastrophic penalty if they fail.
>
> > Q1: How do results change with different α and β values?
>
> Please refer to the `General Response` at the beginning of this rebuttal.
>
> We are more than willing to continue engaging in in-depth discussions. If you have any further questions, please feel free to comment at any time.

---

> ### Author Response · Authors · 2025-11-27
> **Looking Forward to Your Reply**
>
> Thank you once again for taking the time to review our paper and for providing constructive feedback.
>
> We have carefully considered each of your comments and have provided detailed responses. We sincerely hope that our efforts adequately address your concerns and contribute positively to your evaluation.
>
> As the author–reviewer discussion period progresses, we would greatly appreciate any further feedback you may have. If you have any additional questions or require any clarifications, please do not hesitate to reach out to us.

---

> > ### Comment · Reviewer_cqWa · 2025-11-27
> >
> > Thanks for your efforts in addressing my concerns. The theoretical analysis is still too simple. I believe a deeper mathematical analysis would strengthen the motivation of this study.
> >
> > However, I am glad that the authors have added cross-domain generalization results, which demonstrate the strong potential of the proposed algorithm. Accordingly, I would like to increase my score and lower my confidence.

---

### Official Review · Reviewer_6jZE · 2025-10-23

**Soundness:** 2
**Presentation:** 2
**Contribution:** 2
**Rating:** 4
**Confidence:** 3

**Summary:**

This paper addresses the limitations of prevailing algorithms such as GRPO in RLVR, which broadcast a uniform advantage signal across all tokens in a sequence. This coarse-grained approach overlooks the pivotal role of uncertain, high-stakes decisions during reasoning, leading to inefficient exploration and the well-documented problem of entropy collapse. To tackle this, the authors introduce UnCertainty-aware Advantage Shaping (UCAS), a model-free method that refines credit assignment by leveraging the model’s internal uncertainty signals. UCAS operates in two stages: it first modulates the response-level advantage using the model’s overall self-confidence, and then applies a token-level penalty based on raw logit certainty.

**Strengths:**

1.  The topic of this paper is important, as it addresses both the exploration-exploitation trade-off in LLM reasoning and the problem that GRPO broadcast a uniform advantage signal across all tokens in a sequence.
2.  The authors conduct comparisons with up-to-date baselines and report various metrics, including pass@k.
3.  The proposed method is intuitive and makes sense to me.

**Weaknesses:**

1.  The proposed method is mostly based on heuristics; more theoretical analysis should be included.
2.  The idea of response-level and token-level advantage shaping using uncertainty/confidence has already been proposed in works such as Seed-GRPO and Entropy Advantage shaping. I encourage the authors to elaborate more on the unique contributions of their work.
3.  The proposed method may work for GRPO, where the advantage is uniform for all tokens in a trajectory. However, it is unclear how it would apply to PPO.

**Questions:**

See weakness.

---

> ### Author Response · Authors · 2025-11-21
> **To Reviewer 6jZE**
>
> Sincerely thanks for your insightful comments. We appreciate the opportunity to clarify the theoretical underpinnings of our method and its distinct position within the related work.
>
> > **W1:** The method appears heuristic, and the paper needs more theoretical analysis.
>
> We respectfully submit that UCAS is not merely heuristic but a **principled decomposition** of the advantage function designed to address specific theoretical deficiencies in GRPO.
>
> At the response level, our modulation term is derived from the exploration–exploitation imbalance induced by GRPO’s uniform advantage broadcast.  At the token level, our shaping term directly follows from the entropy-collapse mechanism identified in prior RLVR theory ([1]), where a few high-logit tokens dominate the policy gradient.
>
> Thus, UCAS is a theoretically motivated advantage decomposition that uses uncertainty as a fine-grained credit signal, rather than an ad-hoc heuristic.  In addition, we provide strong empirical support via ablations (`Table 2 of the manuscript`) and training dynamics analysis (`Figure 3 of the manuscript`) that isolates and demonstrates the causal effect of each component.
>
> > **W2:** More comparisons with Seed-grpo and Entropy-based shaping methods
>
> We thank the reviewer for this crucial comparison. While the *metrics* (uncertainty/entropy) are shared, the **mechanism and philosophy** of UCAS are fundamentally different, and in some cases, opposite to these works.
>
> - **Difference from Seed-grpo:** Seed-grpo uses the whole response-level semantic entropy to reduce the update scale of uncertain questions (" more conservative update for high uncertainty questions "). This strategy performed in the whole response-level semantic space ignores the emergent role of each key token in the reasoning process, and cannot provide fine-grained guidance for the local reasoning process. In stark contrast, UCAS amplifies the reward for correct uncertain responses (" amplify the reward for correct but uncertain responses "), using response-level signals to guide exploration and token-level signals to regularize local certainty. Models are explicitly encouraged to venture into uncertain domains if they achieve success, whereas Seed-grpo inhibits learning from uncertainty.
> - Recent entropy-based shaping methods often add entropy as a reward bonus. UCAS is unique because it introduces intrinsic epistemic signals such as KL-based **self-confidence** and raw **logit-based penalty** between response-level and token-level. We found that raw logits are a more sensitive proxy for "local overconfidence" than post-softmax entropy, which can be poorly calibrated. This two-stage advantage decomposition mechanism uses response-level signals to guide exploration, while token-level signals regularize local certainty. Furthermore, UCAS introduce a modulation related to correctness (amplifying the penalty for belief errors), which is not present in the pure entropy-based shaping framework. `Table 1 of the manuscript` shows a detailed model performance comparison between our framework and the entropy-based method.
>
> For a more fair exposition of our contribution, we will add comparisons in the revised version.
>
> > **W3:** It is unclear how it would apply to PPO.
>
> UCAS is inherently compatible with PPO as it essentially serves as a "plug-and-play" advantage modifier. In PPO, where the advantage $\hat{A}^{GAE}$ is typically estimated via Generalized Advantage Estimation (GAE) using a value function, UCAS does not replace this process but rather **further calibrates** it. Specifically, UCAS can be applied as a post-processing step to the GAE estimates
>
> We can apply our uncertainty-aware modulation (`Eq. 7 of the manuscript`) and token-level penalty (`Eq. 9 of the manuscript`) directly to $\hat{A}^{GAE}$. This injects the model's intrinsic epistemic uncertainty into the extrinsic value estimate, distinguishing between "confident" and "uncertain" advantages even within the GAE framework.
>
> We hope the above clarifications have enhanced your understanding of our work. We appreciate your valuable feedback and are eager to discuss any additional questions or comments further.
>
> ------
> [1] Cui G, Zhang Y, et al. "The entropy mechanism of reinforcement learning for reasoning language models". arXiv preprint arXiv:2505.22617, 2025.

---

> > ### Comment · Reviewer_6jZE · 2025-11-26
> >
> > Thanks for the authors' response. I am still looking for the additional comparisons and updated elaborations in the revised paper.

---

> > > ### Author Response · Authors · 2025-11-26
> > >
> > > Thank you for your confirmation! We truly appreciate your feedback, which is invaluable for improving our paper.
> > >
> > > We have revised the manuscript and the **updated changes have been marked** within the revision.  We sincerely hope that our efforts adequately address your concerns and contribute positively to your evaluation. Welcome any further questions or comments to help improve our work. **Have a nice day! :)**

---

> ### Comment · Reviewer_6jZE · 2025-11-28
>
> Thanks for the authors' response. I will update my review accordingly (see the following box, as I cannot edit my original review right now).

---

> > ### Comment · Reviewer_6jZE · 2025-11-28
> >
> > **NOTE:** I suddenly found that the edit button for my original review does not work right now, so I cannot update my original review box. Therefore, I have decided to show my final score in this box to let others know: **Rating: 6**. Please inform me if the review box can be edited in the near future.

---

> > > ### Author Response · Authors · 2025-11-28
> > > **Thank you for raising the score!**
> > >
> > > Dear Reviewer 6jZE,
> > >
> > > Thank you for your effort to raise the score, even though it could not be accomplished due to system settings! We sincerely appreciate the valuable feedback you provided, which has been incredibly helpful in improving our work.
> > >
> > > Best,
> > >
> > > Authors

---

### Official Review · Reviewer_L6No · 2025-11-01

**Soundness:** 3
**Presentation:** 3
**Contribution:** 3
**Rating:** 6
**Confidence:** 4

**Summary:**

This paper proposes UCAS, an uncertainty aware advantage shaping method for RLVR training of reasoning LLMs. The idea is to reshape the learning signal in two stages. First, at the response level, the method scales the group normalized advantage by a self confidence score computed as average KL to uniform. Second, at the token level, it subtracts a certainty penalty based on the raw logits for the chosen tokens. Experiments on five math benchmarks and two model sizes show consistent pass@1 gains over GRPO, DAPO, and other recent RLVR variants, with longer reasoning chains and recovery of generation entropy as training proceeds.

**Strengths:**

1. Uncertainty aware shaping that mixes a response level self confidence weight with a token level logit penalty is simple, well motivated, and easy to implement inside existing GRPO or DAPO code. It does not add a new model or a verifier, unlike many PRM based approaches. The method is explained clearly with a compact formula and an algorithm box.
2. On AIME24, MATH 500, AMC, Minerva, and OlympiadBench, UCAS wins on both 1.5B and 7B Qwen math models, with gains over DAPO, KTAE, and Oat Zero. The ablation in Table 2 shows both response level and token level parts contribute, and their combination is best.
3. The work tackles the widely reported entropy collapse in RLVR and shows recovery of generation entropy during training together with longer responses.

**Weaknesses:**

1. All experiments are in math with verifiable final answers. It is unclear if the same shaping works for code unit tests or symbolic tasks, and especially for non binary or dense reward settings.
2. The response level confidence is KL to uniform and the token level proxy is raw logits. Both are known to be imperfect confidence measures and can be miscalibrated.
3. The training uses 16 rollouts per prompt and drops KL and entropy regularizers in some baselines. The paper should include a sensitivity study for alpha and beta, and report results when baselines are tuned with their recommended regularization settings. Otherwise, part of the gain may come from different regularization or longer responses.
4. KTAE [1] also produces token level advantages without extra models. Recent entropy induced advantage methods also reshape the advantage. The novelty margin would be clearer with side by side plots of entropy and pass@k against those methods under the same compute. [1] KTAE: A Model-Free Algorithm to Key-Tokens Advantage Estimation in Mathematical Reasoning

**Questions:**

See weaknesses.

---

> ### Author Response · Authors · 2025-11-21
> **To Reviewer L6No**
>
> Thank you for your thorough review and constructive feedback. We will address your concerns point by point.
>
> > W1: Generalizability beyond math.
>
> Thank you for this comment. To better assess the generality of UCAS beyond mathematical reasoning, we have added a series of experiments on non-math benchmarks, including Leetcode, LivecodeBench, and MMLU, which cover code generation and general reasoning tasks.
>
> **Table L1: Generalizing UCAS from math-only training to evaluations on non-math tasks**
>
> | **Method**      | **LeetCode (Pass@1)** | **LiveCodeBench (Pass@1)** | **MMLU (Acc)**  | **Avg**         |
> | --------------- | --------------------- | -------------------------- | --------------- | --------------- |
> | **Base Model**  | 11.7                  | 5.7                        | 65.7            | 27.7            |
> | **DAPO**        | 18.3                  | 9.2                        | 67.3            | 31.6            |
> | **UCAS (Ours)** | **23.6** (+5.3)       | **14.8** (+5.6)            | **70.8** (+3.5) | **36.4** (+4.8) |
>
> As shown in `Table L1`, we find that, even solely trained on mathematical reasoning data, UCAS consistently outperforms DAPO on these non-math benchmarks. This suggests that the exploration mechanism for uncertainty is a broadly applicable approach that also has gains for the multi-step logic required in programming tasks, as opposed to methods limited to mathematical reasoning. We will include these results in the revised version to demonstrate the broad applicability of our method.
>
> > W2: Uncertainty measures (KL/logits) are imperfect
>
> Our choice of these proxies was a deliberate design trade-off. We aimed for a **model-free** and computationally efficient method that leverages the model's **intrinsic signals** , avoiding the significant cost and complexity of auxiliary networks or model ensembles.
>
> We also specifically chose *raw logits* for the token-level penalty, as prior work ([1]) suggests they can be more direct and less prone to post-softmax calibration issues. As we state in Appendix A, exploring alternative uncertainty metrics is a promising future direction. However, the strong empirical success of our lightweight proxies (`Tables 1 & 2 of the manuscript`)  validates their practical effectiveness for this task.
>
> > W3: Sensitivity of hyperparameters and baseline settings.
>
> About the **sensitivity analysis of hyperparameters**, please refer to the `General Response` at the beginning of this rebuttal.
>
> We followed the setup from the VERL framework and DAPO, which often omit KL/entropy regularizers to isolate the policy optimization algorithm. Our core comparison is **DAPO (baseline) vs. DAPO + UCAS (our method)**, where the only change is our advantage shaping. The gains (e.g., +6.2 on 7B ) are thus directly attributable to UCAS. The fact that UCAS recovers generation entropy (`Figure 3 of the manuscript`) without an explicit entropy regularizer is a key strength, demonstrating that our uncertainty-based shaping inherently promotes the necessary diversity.
>
> > W4: Novelty vs. KTAE and entropy-based methods.
>
> We thank the reviewer for pushing for this clarification.
>
> - vs. KTAE: KTAE is also model-free but uses statistical association (e.g., Fisher Test) to identify key tokens after rollouts. UCAS is fundamentally different. It uses the model's intrinsic epistemic state (uncertainty) during the forward pass. UCAS shapes the advantage based on confidence, whereas KTAE shapes it based on attribution.
> - vs. Entropy Adv: Methods like [2] typically add an entropy term to the advantage to globally encourage high entropy. This can lead to un-productive, random exploration. UCAS is far more targeted: it **does not just reward all entropy**. It selectively rewards uncertain paths that lead to correct answers (Stage 1) and penalizes local overconfidence (Stage 2). This dual-mechanism shaping is a core novel contribution. Our superior results over "GRPO with Entropy Adv." in `Table 1 of the manuscript` support this.
>
> For a more fair exposition of our contribution, we will add comparisons in the revised version.
>
> [1] Huan M, Jing C, et al. "Estimating LLM Uncertainty with Evidence". arXiv preprint arXiv:2502.00290, 2025.
>
> [2] Cheng D, Huang S, et al. "Reasoning with exploration: An entropy perspective". arXiv preprint arXiv:2506.14758, 2025.

---

> ### Author Response · Authors · 2025-11-27
> **Looking Forward to Your Reply**
>
> Thank you once again for taking the time to review our paper and for providing constructive feedback.
>
> We have carefully considered each of your comments and have provided detailed responses. We sincerely hope that our efforts adequately address your concerns and contribute positively to your evaluation.
>
> As the author–reviewer discussion period progresses, we would greatly appreciate any further feedback you may have. If you have any additional questions or require any clarifications, please do not hesitate to reach out to us.

---

### Official Review · Reviewer_EgW8 · 2025-11-01

**Soundness:** 4
**Presentation:** 4
**Contribution:** 3
**Rating:** 6
**Confidence:** 4

**Summary:**

This paper proposes UCAS, a method to improve RLVR for reasoning LLMs by addressing the coarse credit assignment problem in GRPO. Instead of broadcasting a uniform advantage across all tokens, UCAS reshapes the learning signal using the model’s intrinsic uncertainty at two levels: (1) a response-level modulation that amplifies rewards for correct but uncertain trajectories and penalizes overconfident wrong ones, and (2) a token-level certainty penalty that discourages local overconfidence to prevent entropy collapse. Experiments on five mathematical reasoning benchmarks show consistent and significant improvements across both 1.5B and 7B Qwen-Math models, leading to deeper reasoning chains, higher rewards, and better exploration diversity.

**Strengths:**

1. The paper is easy to read and well structured, making the method and motivation intuitive to follow. Improving exploration in RL for reasoning LLMs is a very active and timely topic, and this method provides a valuable and well-motivated contribution in that direction.

2. The method is lightweight, model-free, and compatible with existing RLVR pipelines, which makes it practically useful for scaling to larger models.

3. The experiments are comprehensive, comparing against a wide range of strong RLVR and reasoning baselines with clear and consistent performance gains.

**Weaknesses:**

1. The theoretical foundation of UCAS is mostly intuitive and the paper doesn’t formally analyze why the two-stage shaping leads to more stable optimization or guarantees improved exploration.

2. From Table 2, the token-level certainty component seems to contribute little or even slightly hurt performance in some cases, suggesting its effect is weaker or less stable than the response-level shaping.

3. The paper lacks ablation or sensitivity analysis for the two key hyperparameters \alpha and \beta, which directly control the strength of response-level modulation and token-level penalty. It is unclear how stable UCAS is to different settings of these values.

**Questions:**

1. How do you choose the hypeparameters and how they affect the performance?

2. Since UCAS is conceptually orthogonal to the underlying policy optimization algorithm, have the authors tested it with other GRPO-family methods (e.g., DAPO or GSPO)? Demonstrating consistent gains across multiple RLVR optimizers would strengthen the claim that UCAS provides generalizable advantage shaping rather than optimizer-specific benefits.

---

> ### Author Response · Authors · 2025-11-21
> **To Reviewer EgW8**
>
> Thanks for your insightful comments and we believe they hold significant value for our work. We try to resolve your concerns below.
>
> > W1: Theoretical analysis on why UCAS is effective
>
> Thank you for this feedback. We need to clarify that our method is not arbitrary heuristic. It is principled on the well-established idea of balancing exploration-exploitation trade-off.
>
> Our two-stage mechanism is a direct implementation of this principle: 1.  Stage 1 (Response-Level) explicitly rewards valuable exploration (correct-but-uncertain paths) and penalizes over-exploitation (incorrect-but-confident paths). 2.  Stage 2 (Token-Level) acts as a local regularizer to prevent premature convergence by penalizing overconfidence.
>
> We back this principled design with extensive empirical evidence (`Tables 1 & 2 of the manuscript`) and detailed analysis of the training dynamics (`Figure 3 of the manuscript`) , which demonstrate its practical effectiveness.
>
> > W2: Token-level contribution seems weak or unstable.
>
> We thank the reviewer for this sharp observation, which prompts us to clarify the results of `Table 2 in the manuscript`. The token-level penalty provides a consistent and crucial benefit, and its complementarity with the response-level component is key to UCAS.
>
> - For the **1.5B model**, the token-level penalty alone ("w/ DAPO + Token-Level Certainty") achieved an average score of 45.1, which was actually a *stronger* individual gain (+3.9) than the response-level modulation alone (44.7, +3.5 gain).
> - For the **7B model**, it provided a solid +2.2 gain (52.7 vs 50.5).
>
> Most importantly, in *both* 1.5B and 7B models, the **full UCAS** (combining both components) achieves the highest score (47.3 and 56.7, respectively). This confirms our claim that the two signals are "jointly necessary". The token-level penalty's main role is to preserve local diversity and prevent the premature entropy collapse shown in `Figure 3 of the manuscript`, which enables the strategic response-level modulation to work more effectively over a longer training period.
>
> > W3&Q1: Sensitivity analysis of the two hyperparameters α and β
>
> Please refer to the `General Response` at the beginning of this rebuttal.
>
> > **Q2: How effective is the integration of UCAS with other optimization algorithms？**
>
> Thanks for your comments. Actually, our UCAS implementation is built directly on top of DAPO. As shown in our objective function (`Eq. 10 of the manuscript`), we adopt the decoupled clipping bounds ($1-\epsilon_{low}, 1+\epsilon_{high}$) from DAPO. Furthermore, our primary baseline for all comparisons in `Table 1 of the manuscript` and the ablation study in `Table 2 of the manuscript`  is DAPO. The consistent and significant gains we report (e.g., +6.1 on 1.5B, +6.2 on 7B) are *directly over* a strong DAPO baseline. This demonstrates that UCAS provides a generalizable advantage shaping mechanism that effectively enhances modern RLVR optimizers like DAPO.

---

> > ### Author Response · Authors · 2025-11-27
> > **Looking Forward to Your Reply**
> >
> > Thank you once again for taking the time to review our paper and for providing constructive feedback.
> >
> > We have carefully considered each of your comments and have provided detailed responses. We sincerely hope that our efforts adequately address your concerns and contribute positively to your evaluation.
> >
> > As the author–reviewer discussion period progresses, we would greatly appreciate any further feedback you may have. If you have any additional questions or require any clarifications, please do not hesitate to reach out to us.

---

### Author Response · Authors · 2025-11-21
**General Response about the sensitivity analysis of hyperparameters**

Thank reviewer `EgW8`,  `L6No` and `cqWa` for their constructive feedback regarding the sensitivity of our method to the hyperparameters $\alpha$ (response-level modulation) and $\beta$ (token-level penalty). We greatly appreciate your attention to this matter, and below, we address this issue from three perspectives to ensure the reliability of our findings.  We analyzed the impact of each hyperparameter by varying one while fixing the other at its optimal value.

**Table G1: Performance comparison varying Token-Level Penalty ($\beta$)**

| Response-Level $\alpha$ | Token-Level $\beta$ | Math-500 |
| :---------------------- | ------------------- | :------- |
| 0.2                     | 0.005               | 79.6     |
| 0.2                     | 0.01                | 80.4     |
| 0.2                     | 0.05                | 78.8     |
| 0.2                     | 0.1                 | 78.4     |

**Table G2: Performance comparison varying Response-Level Modulation ($\alpha$)**

| Response-Level $\alpha$ | Token-Level $\beta$ | Math-500 |
| :---------------------- | ------------------- | :------- |
| 0.1                     | 0.01                | 80.0     |
| 0.2                     | 0.01                | 80.4     |
| 0.4                     | 0.01                | 78.8     |

1. **Impact of Token-Level Penalty ($\beta$)** : Table G1 illustrates the performance changes when varying the token-level penalty coefficient $\beta$, with the response-level modulation fixed at $\alpha=0.2$. A moderate $\beta$ (0.01) effectively mitigates entropy collapse, improving reasoning stability. However, as $\beta$ increases beyond 0.05, performance begins to degrade. This suggests that an excessive penalty may over-regularize the model, stifling the necessary exploration during the generation process.

2. **Impact of Response-Level Modulation ($\alpha$)** : Table G2 demonstrates the sensitivity to the response-level modulation coefficient $\alpha$, with the token-level penalty fixed at $\beta=0.01$. The results indicate that $\alpha$ is relatively stable within the range of $[0.1, 0.2]$. Increasing $\alpha$ to 0.4 leads to a performance drop, implying that if the uncertainty modulation is too strong, it may dominate the reward signal, interfering with the advantage estimation's accuracy.

3. **Hyperparameter Selection Strategy**: In response to Reviewer `EgW8's` question regarding how these values are chosen:

   Our parameter selection is grounded in our theoretical framework of **hierarchical dominance adjustment**. Functionally, $\alpha$ and $\beta$ serve to scale the value ranges of the response-level uncertainty and token-level logits, respectively.

   Therefore, rather than relying on random search, we select these hyperparameters based on the **magnitude of the model's output signals**. The goal is to align the scales of the uncertainty penalties with the model's advantage estimates, ensuring an appropriate **cooperative adjustment effect** where the uncertainty signal guides exploration without overwhelming the primary learning objective. As demonstrated by the sensitivity analysis above, this theoretically guided selection strategy aligns well with the empirically optimal region.

---

### Author Response · Authors · 2025-11-26
**Response and Updated Manuscript to All Reviewers**

**We sincerely thank all reviewers (`EgW8`, `L6No`, `6jZE`,`cqWa`) for their thoughtful and constructive feedback.** We are encouraged that reviewers recognized the simplicity and effectiveness of our method, the importance of the problem (addressing entropy collapse in RLVR), and our strong empirical results across multiple benchmarks.

Your insightful comments have driven substantial improvements to our work. In response, we have extensively revised the manuscript (**changes marked in the revision**). Below is a summary of the key updates:
- **Sensitivity Analysis** (Appendix B): Addressing concerns from Reviewers `EgW8`, `L6No`, and `cqWa`, we added a comprehensive ablation study on hyperparameters $\alpha$ and $\beta$, demonstrating the method's robustness.
- **Generalizability Analysis** (Analysis Section): In response to `L6No`, we added experiments on LeetCode, LiveCodeBench, and MMLU, proving UCAS's effectiveness extends to code generation and general reasoning.
- **Clarified Novelty & Comparisons** (Related Work): Addressing `6jZE` and `cqWa`, we revised the Related Work section to explicitly contrast UCAS’s exploratory philosophy with Seed-GRPO’s conservative approach and Entropy-based shaping method.
- **Theoretical Grounding**: We refined the methodological description to clarify the signal-to-noise motivation behind the exponential weighting and the conditional logic of our advantage shaping.

We believe these updates further justify our approach to addressing critical challenges in LLM Reasoning. We hope the revised submission meets your expectations and demonstrates the value of our contributions.

Thanks again for your constructive feedback and support!

---

### Note · Authors · 2025-12-08

**Comment:**

Thank you to all the reviewers for your thoughtful and constructive feedback.  After careful consideration, we acknowledge that the current version of this work still has notable gaps that prevent it from meeting the standards expected for publication.

We plan to thoroughly address these concerns and significantly improve the work before considering resubmission elsewhere.  Thank you again for your valuable input.

**Withdrawal Confirmation:**

I have read and agree with the venue's withdrawal policy on behalf of myself and my co-authors.